# Cytokine Response of Natural Killer Cells to Hepatitis B Virus Infection Depends on Monocyte Co-Stimulation

**DOI:** 10.3390/v16050741

**Published:** 2024-05-08

**Authors:** Paul Kupke, Johanna Brucker, Jochen M. Wettengel, Ulrike Protzer, Jürgen J. Wenzel, Hans J. Schlitt, Edward K. Geissler, Jens M. Werner

**Affiliations:** 1Department of Surgery, University Hospital Regensburg, 93053 Regensburg, Germany; 2Institute of Virology, School of Medicine and Health/Helmholtz Munich, Technical University of Munich, 81675 Munich, Germany; 3German Center for Infection Research (DZIF), Munich Partner Site, 81675 Munich, Germany; 4Institute of Clinical Microbiology and Hygiene, University Hospital Regensburg, 93053 Regensburg, Germany

**Keywords:** hepatitis B virus, HBV, natural killer cells, NK cells, monocytes, bystander activation, HepG2, HepaRG, NTCP

## Abstract

Hepatitis B virus (HBV) is a major driver of chronic hepatic inflammation, which regularly leads to liver cirrhosis or hepatocellular carcinoma. Immediate innate immune cell response is crucial for the rapid clearance of the infection. Here, natural killer (NK) cells play a pivotal role in direct cytotoxicity and the secretion of antiviral cytokines as well as regulatory function. The aim of this study was to further elucidate NK cell responses triggered by an HBV infection. Therefore, we optimized HBV in vitro models that reliably stimulate NK cells using hepatocyte-like HepG2 cells expressing the Na^+^-taurocholate co-transporting polypeptide (NTCP) and HepaRG cells. Immune cells were acquired from healthy platelet donors. Initially, HepG2-NTCP cells demonstrated higher viral replication compared to HepaRG cells. Co-cultures with immune cells revealed increased production of interferon-γ and tumor necrosis factor-α by NK cells, which was no longer evident in isolated NK cells. Likewise, the depletion of monocytes and spatial separation from target cells led to the absence of the antiviral cytokine production of NK cells. Eventually, the combined co-culture of isolated NK cells and monocytes led to a sufficient cytokine response of NK cells, which was also apparent when communication between the two immune cell subpopulations was restricted to soluble factors. In summary, our study demonstrates antiviral cytokine production by NK cells in response to HBV^+^ HepG2-NTCP cells, which is dependent on monocyte bystander activation.

## 1. Introduction

Hepatitis B virus (HBV) infection is currently one of the leading causes of acute as well as chronic viral hepatitis and is a major driver of liver cirrhosis and liver cancer, which contributes significantly to liver-related mortality globally. Frequently, liver transplantation becomes necessary. Therefore, HBV infections represent a serious burden for healthcare systems worldwide [1,2].

HBV primarily replicates in hepatocytes and can persist for a long time despite constantly improving treatment options. The interplay of innate and adaptive immune responses plays a key role in the defense against HBV and is decisive for the course of the infection [3,4]. Viral infection usually leads to a rapid immune response of the innate immune system in order to prevent the persistence of the virus and to stimulate further antiviral mechanisms [5,6]. The control of HBV infection is therefore often successful, but the complete elimination of the virus remains a major challenge [7].

Natural killer (NK) cells are a key element of the first line of defense. Through a rapid immune response, they are substantially involved in HBV infection, not only through direct cytotoxicity against infected cells but also through the massive release of antiviral cytokines [8]. The main cytokines in the immune response of NK cells to viral pathogens are interferon (IFN)-γ and tumor necrosis factor (TNF)-α, with IFN-γ in particular playing a key role in the control of HBV infections [7,8]. Furthermore, NK cells are involved in the coordination and priming of other immune cell populations. In chronic HBV-infected patients, the production of cytokines by NK cells is impaired [8].

Various NK cell subpopulations have been identified. NK cells are most frequently classified according to the expression of CD56 and CD16. CD56^dim^/CD16^+^ NK cells normally exhibit cytotoxic functions, while CD56^bright^/CD16^−^ NK cells are primarily relevant for regulatory functions and the production of cytokines [9].

The production of cytokines by NK cells regularly follows a stimulation by further immune cell populations, for example, with interleukin (IL)-12, IL-15, and IL-18, which is commonly adapted in ex vivo/in vitro studies to detect reasonable cytokine responses by NK cells [9,10].

In this study, we investigated the role of NK cells in the acute phase of an in vitro HBV infection and focused on further immune cell populations that are necessary for NK cells to exert their full antiviral capacity.

## 2. Materials and Methods

### 2.1. HBV Culture Systems

For this study, the two human hepatoma cell lines HepG2-NTCP (Prof. Ulrike Protzer, Munich, Germany) and HepaRG (Biopredic International, Saint-Grégoire, France) were used. Both express the Na^+^-taurocholate co-transporting polypeptide (NTCP), which is one receptor of HBV to enter hepatocytes by endocytosis [7]. The cells were cultivated in T75 flasks and for experiments in 12-well plates (both TPP, Trasadingen, Switzerland) at 2 × 10^5^ cells/mL in William’s E medium (Fisher Scientific, Hampton, NH, USA), supplemented with 10% HyClone FetalClone II Serum (Fisher Scientific), 1% penicillin/streptomycin (Sigma-Aldrich, St. Louis, MO, USA), 1% L-glutamine (Sigma-Aldrich), 0.023 IE/mL insulin (Sanofi, Paris, France), 4.7 µg/mL hydrocortisone (Pfizer, New York, NY, USA), and 80 µg/mL gentamicin (Ratiopharm, Ulm, Germany). The medium for HepG2-NTCP cells included 10 µg/mL Blasticidin S (Invitrogen, Waltham, MA, USA). All cultures were performed at 37 °C and 5% CO_2_. HepG2-NTCP cells were cultured 72 h before inoculation and had 1.8% dimethyl sulfoxide (DMSO, AppliChem, Darmstadt, Germany) supplemented during the last 48 h, whereas HepaRG had to follow a differentiation protocol for four weeks with 1.8% DMSO added during the last two weeks to express the NTCP receptor [10,11]. The medium was renewed twice a week. Inoculation with a highly purified HBV (genotype D, serotype ayw) [12] was then performed at a multiplicity of infection (MOI) of 200 with a growth medium containing 4% polyethylene glycol 8000 (Promega, Madison, WI, USA).

### 2.2. Co-Culture with Immune Cells

Mononuclear cells of the peripheral blood (PBMCs) were obtained from healthy platelet donors as a byproduct of apheresis and then isolated by density gradient centrifugation. The PBMCs from one donor were then processed individually. Each donor was tested negative for HBV by serology and NAT. For co-culture, PBMCs were added at an effector-to-target ratio of 1:1 in RPMI 1640 + GlutaMAX (Fisher Scientific), supplemented with 10% HyClone FetalClone II Serum (Fisher Scientific), 1% sodium pyruvate (Fisher Scientific), 1% MEM nonessential amino acids (Fisher Scientific), 7.5% sodium bicarbonate (Fisher Scientific), 1% penicillin/streptomycin (Sigma-Aldrich), and 50 µM 2-mercaptoethanol (Sigma-Aldrich). The proportion of NK cells within the PBMCs of each donor was assessed by flow cytometry before performing experiments using isolated NK cells to determine the optimal cell count for the co-culture. Finally, the absolute NK cell count was equal for every condition of each donor. Co-culture was performed for 24 h on day 14 after infection, as it was demonstrated that the proportion of infected target cells reached roughly 40% at this time point [13]. Where indicated, transwells (0.4 µm, Corning, Corning, NY, USA) were used to achieve spatial separation between the cell populations, only allowing for the exchange of diluted factors.

To isolate or deplete unlabeled immune cell subpopulations, PBMCs were treated with NK Cell or Pan Monocyte Isolation Kits (Miltenyi Biotec, Bergisch Gladbach, Germany) or the corresponding CD56 or CD14 MicroBeads (Miltenyi Biotec) followed by separation with the autoMACS Pro Separator (Miltenyi Biotec) according to the manufacturer’s protocol.

### 2.3. HBV DNA Quantification

Nucleic acid was extracted on an EZ1 Advanced XL workstation using the EZ1 Virus Mini Kit v2.0 (Qiagen, Hilden, Germany) with 100 μL elution. A 5 μL aliquot of the eluate was used for quantitative PCR (qPCR) using 2× TaqMan Universal PCR Master Mix, no UNG (Applied Biosystems, Darmstadt, Germany). Two replicates were analyzed in 30 μL reactions including specific primers and a TaqMan hydrolysis probe located in the S region of the HBV genome (sAg-F: 5′-caa cct cca atc act cac caa c-3′, sAg-R: 5′-ata tga taa aac gcc gca gac ac-3′, and sAg-So: FAM-5′-tcc tcc aay ttg tcc tgg tta tcg ct-3′-TAMRA). Thermal cycling was carried out on a StepOnePlus instrument (Applied Biosystems) and comprised an initial 10 min 95 °C step, followed by 45 cycles of 95 °C for 15 s and 60 °C for 60 s. The number of cellular genomes in each sample was estimated by qPCR targeting a noncoding chromosomal region upstream of the PDHB gene (human genome assembly hg38, chromosome 3, amplicon position: 58,436,139–58,436,218, 80 bp; PDH-F: 5′-tcg atc ggg act gct ttc c-3′; PDH-R: 5′-ccc aca acc tag cac caa aag a-3′; and PDH-So: FAM-5′-cat ctc ctt ttg ctt ggc aaa tct gat cc-3′-TAMRA). Quantitative HBV PCR results were then expressed as genome copies per million cells (c/10^6^ cells). In order to achieve this, qPCR standard curves were obtained by amplification of a series of defined amounts of pre-characterized, assay-specific recombinant HBV-DNA plasmid standards. The specific target concentrations in the specimens were obtained by comparing the respective quantification cycles (Ct-values) to these standard curves. The number of cellular genomes in each sample was determined in a similar way by a PDHB assay and used to calculate the HBV genome numbers in a standardized way per million cells in each specimen.

### 2.4. HBeAg Detection

Hepatitis B e antigen (HBeAg) was detected from cell culture supernatants on an Architect i1000sr instrument (Abbott, Wiesbaden, Germany) using a diagnostic qualitative chemiluminescent microparticle immunoassay (CMIA, Architect HBeAg Assay, off-label, Abbott, Green Oaks, IL, USA). Qualitative results (detected/not detected) and signal-to-cutoff ratios (S/CO) were recorded for each measurement.

### 2.5. Stimulation of PBMCs Prior to Staining

Immune cells were stimulated prior to flow cytometry staining. As recently described [10], a combination of IL-12 and IL-15 (both R&D Systems, Minneapolis, MN, USA) was used for 24 h at concentrations of 0.5 ng/mL and 20 ng/mL, respectively. During the last 4 h, protein transport inhibitors containing brefeldin A and monensin (both BD Biosciences, Franklin Lakes, NJ, USA) were added.

### 2.6. Flow Cytometry

All analyses were performed on a FACSCanto II flow cytometer using FACSDiva software v6 and FlowJo v10 (all BD Biosciences).

First, dead cells were excluded by staining with ethidium monoazide bromide (Sigma-Aldrich) for 10 min on ice under direct light. NK cells were then identified after the exclusion of monocytes and B cells by CD14 and CD19 as CD56^+^/CD3^−^ lymphocytes (see Appendix A). Surface staining and intracellular staining were each performed for 30 min at 4 °C in the dark. Before the intracellular staining, cells were fixed and permeabilized with the Cytofix/Cytoperm kit (BD Biosciences).

The following antibodies were used for flow cytometry staining: anti-CD14-PerCP-Cy5.5 (BD Biosciences), anti-CD19-PerCP-Cy5.5 (BD Biosciences), anti-CD3-APC-Cy7 (BioLegend, San Diego, CA, USA), anti-CD56-PE-Cy7 (BD Biosciences), anti-IFN-γ-APC (BD Biosciences), and anti-TNF-α-V450 (BD Biosciences).

### 2.7. Statistics

Analyses were performed using GraphPad Prism v10 (GraphPad Software, La Jolla, CA, USA). Statistical tests were used as indicated in the respective figure legends; *p*-values < 0.05 were considered statistically significant.

## 3. Results

### 3.1. Comparison of Different Cell Lines on HBV Replication and the Effect of Co-Culture with PBMCs

To establish the optimal conditions for HBV replication, we compared two of the most commonly used hepatoma cell lines [14] concerning viral replication and HBeAg production. Here, HepG2-NTCP cells, compared to HepaRG cells, showed higher concentrations of HBV DNA (2.4 × 10^7^ compared to 3.3 × 10^6^ c/10^6^ cells and 4.3 × 10^7^ compared to 8.0 × 10^6^ c/10^6^ cells) and HBeAg in the culture supernatants (23.49 compared to 4.91 S/CO and 25.10 compared to 3.74 S/CO) 13 and 19 days after inoculation, respectively (see Figure 1A).

As shown in Figure 1B, the addition of PBMCs to the HBV culture system led to a decrease in HBeAg in the culture supernatants (6.38 compared to 4.92 S/CO, *p* = 0.0230). The co-culture of isolated CD56^+^ NK cells led to a similar HBeAg level to that obtained without the addition of any immune cells (6.38 compared to 6.32 S/CO, *p* > 0.9999).

### 3.2. Production of Antiviral Cytokines by NK Cells after Contact with HBV

After 24 h of the co-culture of PBMCs with HepG2-NTCP cells, as shown in Figure 2A,B, HBV^+^ target cells led to an increase in IFN-γ production in NK cells (median 11.17 [IQR 7.46–21.02] compared to 9.86% [IQR 2.66–15.12] IFN-γ^+^ NK cells, *p* < 0.0001; MFI 4472 [IQR 736–7651] compared to 3431 [IQR 714–6472], *p* = 0.0002). When comparing CD56^bright^ to CD56^dim^ NK cells (see Figure 2C), the mainly cytokine-producing CD56^bright^ NK cells revealed a larger increase (9.81 compared to 2.62% change in IFN-γ^+^ NK cells, *p* = 0.0015). Cultivation with HBV without target cells did not reveal any stimulation (see Appendix A).

As seen in Figure 2D,E, after contact with HBV^+^ HepG2-NTCP cells, the production of TNF-α was also increased (median 1.44 [IQR 0.80–3.94] compared to 0.98% [IQR 0.64–3.73] TNF-α^+^ NK cells, *p* = 0.0005; MFI 623 [IQR 252–1244] compared to 459 [IQR 173–877], *p* = 0.0015). Here, CD56^bright^ and CD56^dim^ NK cells showed a similar increase in TNF-α production (see Figure 2F).

### 3.3. The Antiviral Capacity of Isolated NK Cells

To assess whether NK cells can perform antiviral cytokine production as an isolated cell population without the concomitant stimulation of other immune cell populations, changes in NK cell cytokine production were compared as part of all PBMCs and after magnetic isolation, as shown in Figure 3. After co-culture and compared to NK cells as part of PBMCs, isolated NK cells showed an unaltered level of IFN-γ production as a response to HBV^+^ target cells compared to mock-infected HepG2-NTCP cells (2.84 compared to −1.32% change in IFN-γ^+^ NK cells, *p* = 0.0002). This was also evident after the breakdown into CD56^bright^ (9.81 compared to −5.40% change in IFN-γ^+^ NK cells, *p* = 0.0003) and CD56^dim^ NK cells (2.62 compared to −2.69% change in IFN-γ^+^ NK cells, *p* = 0.0012).

Similar results were evident when analyzing the TNF-α production of isolated NK cells (0.28 compared to −0.02% change in TNF-α^+^ NK cells, *p* = 0.0015), also after separation in CD56^bright^ (0.27 compared to −0.55% change in TNF-α^+^ NK cells, *p* = 0.0020) and CD56^dim^ NK cells (0.30 compared to 0.01% change in TNF-α^+^ NK cells, *p* = 0.0037).

### 3.4. The Role of Monocytes and Spatial Conditions in NK Cell Stimulation

Compared to the co-culture as part of the total PBMC population (24.52% IFN-γ^+^ NK cells and 3.83% TNF-α^+^ NK cells), as shown in Figure 4A,B, the depletion of CD14^+^ monocytes abolished the antiviral effect of NK cells following contact with HBV^+^ HepG2-NTCP cells (13.70 to 6.40% IFN-γ^+^ NK cells, *p* = 0.3629; 1.08 to 0.79% TNF-α^+^ NK cells, *p* = 0.8454).

As shown in Figure 4C,D, compared to the co-culture with direct contact with the target cells (23.17% IFN-γ^+^ NK cells and 6.19% TNF-α^+^ NK cells), the spatial separation of the PBMCs using transwells abolished the antiviral effect of NK cells following contact with HBV^+^ target cells (4.05 to 6.64% IFN-γ^+^ NK cells, *p* = 0.8454; 1.22 to 0.79% TNF-α^+^ NK cells, *p* > 0.9999).

### 3.5. Direct Contact between Monocytes, NK Cells, and HBV^+^ Target Cells

Next, as our data showed a contact- and monocyte-dependent antiviral cytokine response of NK cells, we proceeded to further elucidate the significance of the spatial combination of NK cells and monocytes in relation to the HBV^+^ HepG2-NTCP cells.

Here, as shown in Figure 5, the combination of isolated NK cells and isolated monocytes, both separated from the target cells by transwells, revealed no antiviral cytokine response by NK cells (−0.34% change in IFN-γ^+^ NK cells, *p* = 0.6875; −0.02% change in TNF-α^+^ NK cells, *p* = 0.6875). When monocytes were cultured in direct contact with the target cells with NK cells separated by transwells from both monocytes and target cells, a diminished antiviral cytokine response was observed (1.40% change in IFN-γ^+^ NK cells, *p* = 0.0156; 0.22% change in TNF-α^+^ NK cells, *p* = 0.0156). Without the restriction of cell contact between monocytes, NK cells, and their target cells, NK cells revealed the highest level of antiviral cytokine production (4.77% change in IFN-γ^+^ NK cells, *p* = 0.0469; 0.70% change in TNF-α^+^ NK cells, *p* = 0.0156).

## 4. Discussion

Chronic HBV infection is one of the main causes of liver cirrhosis worldwide. While rapid NK cell response is crucial in the early stages of an HBV infection [15], studies in chronically HBV-infected patients demonstrate exhausted NK cell function that contributes to impaired HBV clearance [16]. In peripheral blood, NK cells account for around 10% of all mononuclear cells [17]; however, the frequency within the liver can rise to 50% [15]. It is important to emphasize that a pronounced NK cell response, in addition to the early clearance of the virus, may contribute to significant liver damage if HBV infection persists [6,18,19,20]. Notably, patients with chronic HBV infections show an altered NK cell phenotype with an impaired antiviral cytokine response, maintained cytotoxicity, and an elevated frequency of TRAIL-expressing CD56^bright^ NK cells [8].

Therefore, a rapid and coordinated response of NK cells is important in order to strive for timely viral clearance and regulatory functionality while preventing cellular exhaustion or excessive immunity with the risk of significant liver damage [8,21,22].

Thus, we optimized an in vitro HBV model with the HepG2-NTCP cell line to reliably stimulate NK cells and investigate the immediate immune response. Here, in accordance with the existing data [7,23], NK cells increased the production of antiviral cytokines IFN-γ and TNF-α. In particular, CD56^bright^ NK cells, which are the predominant cytokine-producing subpopulation [24], showed a sharp increase in IFN-γ in our study. Notably, our data demonstrate that the direct cell contact of PBMCs to the target cells was essential, and isolated NK cells showed no cytokine response to HBV. In the end, monocytes were the responsible bystander population, which led to an unrestricted cytokine response of NK cells. Joint contact with the infected target cells was necessary for the full effect. Interestingly, contact of the monocytes with the target cell was enough to stimulate NK cells in an attenuated form via the cell culture supernatant.

The importance of a pronounced interaction between monocytes and NK cells has been emphasized by previous studies [25,26,27,28,29,30,31], but evidence in the field of HBV infection is still scarce. Li et al. [32] demonstrated an immunosuppressive cascade in which suppressive monocytes generated by HBV initiate the differentiation of regulatory NK cells, which in the end results in T-cell inhibition. Further knowledge about the NK cell–monocyte axis in viral hepatitis was primarily gained in the field of HCV infections, and the findings of our study are consistent with these. Serti et al. [33] demonstrated that an attenuated IFN-γ response by NK cells in chronically HCV-infected patients was not caused by an intrinsic mechanism but rather by a suboptimal co-stimulation by monocytes. The role of accessory cells and the necessity of monocytes for maximal IFN-γ production by NK cells in HCV infection was highlighted by Zhang et al. [34], and a mechanism following a direct cell–cell contact was demonstrated by Pollmann et al. [35]. According to their study, monocytes were necessary to achieve NK cell antiviral functionality in HCV infection. Nishio et al. [36] revealed a monocyte-driven increase in NK cell cytotoxicity in HCV infection, which could eventually contribute to liver injury and the persistence of the infection. Generally, the majority of the studies address the influence of monocytes on NK cells. However, Klöss et al. [37] demonstrated that NK cells affect the response of monocytes in HCV infection, indicating a mutual interaction. The study by Askenase et al. [38] could also prove that NK cells are directly involved in the priming of monocytes during infections.

Limitations of our study are that the HBV infection model is only based on the well-established highly HBV-replicating HepG2-NTCP cell line, and NK cell function was solely assessed by flow cytometry. Additionally, only extracellular viral detection was performed. Further experiments have to confirm our findings in primary human hepatocytes and investigate the exact mechanism by which monocytes stimulate NK cells in the acute setting of an HBV infection and how this interplay suppresses HBV replication to assess the clinical applications of our findings.

## 5. Conclusions

In summary, our in vitro study reveals a contact-dependent stimulation of NK cells by HBV^+^ HepG2-NTCP cells with increased production of antiviral cytokines IFN-γ and TNF-α. For this, monocyte bystander activation is necessary.

## Figures and Tables

**Figure 1 viruses-16-00741-f001:**
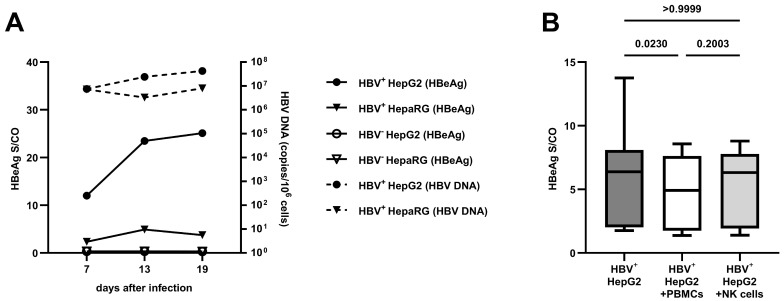
Verification of HBV replication: (**A**) the progression of HBV DNA and HBeAg in the cell culture supernatants of HepG2-NTCP and HepaRG cells as a median of two independent experiments; (**B**) the effect of PBMCs and isolated NK cells on HBeAg after 24 h of co-culture (*n* = 18). Appearance: median with interquartile range (whiskers min to max); statistical analysis: Friedman test with Dunn’s multiple-comparison test.

**Figure 2 viruses-16-00741-f002:**
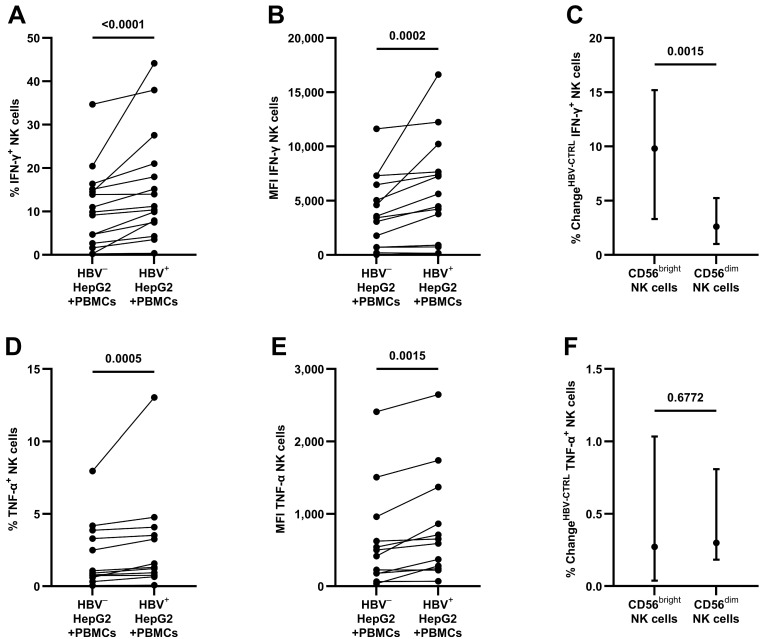
Antiviral cytokine response by NK cells after contact with HBV^+/−^ HepG2-NTCP cells for 24 h: (**A**) the frequency of IFN-γ^+^ NK cells and (**B**) the respective MFI values (*n* = 15); (**C**) the change in frequency (HBV^+^ – HBV^−^) of IFN-γ^+^ NK cells comparing CD56^bright^ and CD56^dim^ NK cells (*n* = 15); (**D**) the frequency of TNF-α^+^ NK cells and (**E**) the respective MFI values (*n* = 12); (**F**) the change in frequency (HBV^+^ – HBV^−^) of TNF-α^+^ NK cells comparing CD56^bright^ and CD56^dim^ NK cells (*n* = 12). Appearance: median with interquartile range (**C**,**F**); statistical analysis: Wilcoxon matched-pair signed-rank test.

**Figure 3 viruses-16-00741-f003:**
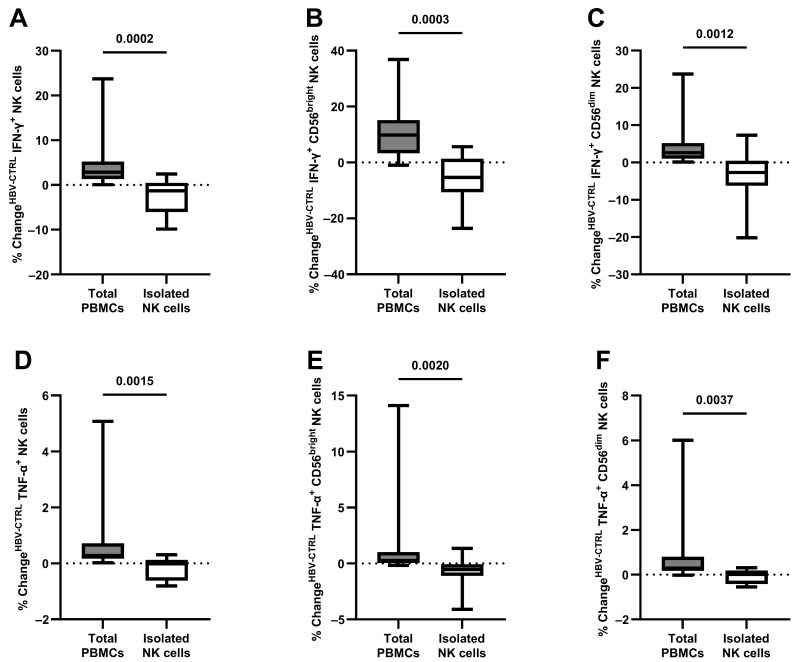
Loss of antiviral cytokine response after isolation of CD56^+^ NK cells; co-culture with HBV^+/−^ HepG2-NTCP cells for 24 h: (**A**) the change in frequency (HBV^+^ – HBV^−^) of isolated IFN-γ^+^ NK cells compared to NK cells within PBMCs, also separated in (**B**) CD56^bright^ NK cells and (**C**) CD56^dim^ NK cells (*n* = 15 and 10); (**D**) the change in frequency (HBV^+^ – HBV^−^) of isolated TNF-α^+^ NK cells compared to NK cells within PBMCs, also separated in (**E**) CD56^bright^ NK cells and (**F**) CD56^dim^ NK cells (*n* = 12 and 10). Appearance: median with interquartile range (whiskers min to max); statistical analysis: Wilcoxon matched-pair signed-rank test.

**Figure 4 viruses-16-00741-f004:**
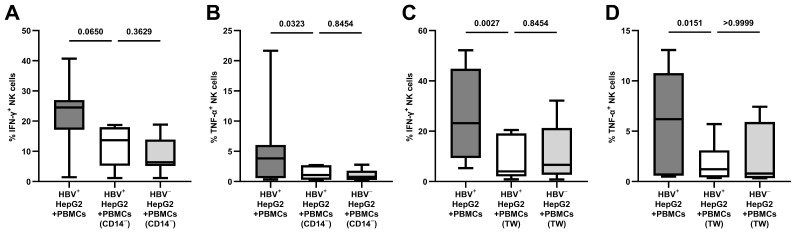
Loss of antiviral cytokine response after depletion of CD14^+^ monocytes and spatial separation by transwells; co-culture with HBV^+/-^ HepG2-NTCP cells for 24 h: (**A**) the frequency of IFN-γ^+^ NK cells following the depletion of CD14^+^ monocytes and (**B**) the respective frequency of TNF-α^+^ NK cells; (**C**) the frequency of IFN-γ^+^ NK cells following spatial separation by transwells and (**D**) the respective frequency of TNF-α^+^ NK cells (*n* = 7). Appearance: median with interquartile range (whiskers min to max); statistical analysis: Friedman test with Dunn’s multiple-comparison test.

**Figure 5 viruses-16-00741-f005:**
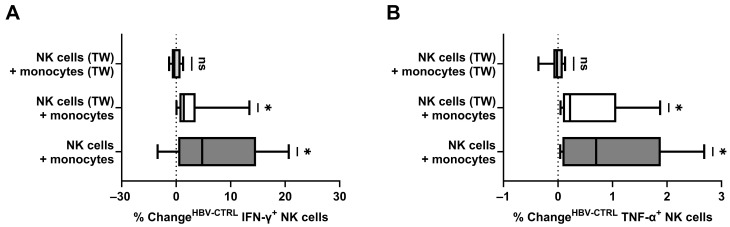
Restored cytokine production of CD56^+^ NK cells after combination with CD14^+^ monocytes depending on the communication possible; co-culture with HBV^+/−^ HepG2-NTCP cells for 24 h: (**A**) the change in frequency (HBV^+^ − HBV^−^) of IFN-γ^+^ NK cells depending on the usage of transwells (TW) and (**B**) the respective frequency of TNF-α^+^ NK cells (*n* = 7). Appearance: median with interquartile range; statistical analysis: one-sample Wilcoxon signed-rank test. * indicates *p* < 0.05.

## Data Availability

The original data presented in the study are available upon request from the corresponding author.

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
