# Peer review of "Cytokine Response of Natural Killer Cells to Hepatitis B Virus Infection Depends on Monocyte Co-Stimulation"

_viruses, 2024, doi:10.3390/v16050741_

Round 1

Reviewer 1 Report

Comments and Suggestions for Authors

The manuscript: "Cytokine response of natural killer cells to hepatitis B virus infection depends on monocyte co-stimulation" has a brief Introduction that, however, contains sufficient information on the subject. From what I understand, both of the sentences in lines 50-52 relate to the reference number 8, so the authors can put the reference number in the text only once, in line 52. Methods are described in detail and Results bring substantial evidence on production of cytokines in NK-cells being dependent on monocytes, figures corroborate the textual data. Discussion part is quite short, considering there are quite a few papers published on the subject and I would advise the authors to elaborate and compare their results with other publications on the subject a bit more. There is also some issues with references in the discussion part, paragraph; lines 252-254 contains no references so authors should either provide one or state that it is their data. Same goes for the lines 266-272, it is not quite clear whether they are referring to the reference number 30 or their own data. Other than that I find the manuscript suitable for publication.

Comments on the Quality of English Language

The English language requires only minor interventions that can be resolved through editing process. 

Reviewer 2 Report

Comments and Suggestions for Authors

This manuscript describes the NK cell response induced by HBV infection using HepG2-NTCP cell system. This study may be useful to understand the immune response against HBV infection. However, there are several concerns. Results did not fully support the conclusion. Additionally, Materials and Methods are not adequately described, making it difficult to evaluate this manuscript.

Major Comments

1. Is the presence of HBV-infected cells necessary for the production of interferon gamma and TNF alpha by NK cells? Is there any change in NK cells when mixing only PBMCs and HBV?

2. I consider primary human hepatocytes also should be used in some, but not all, experiments.

3. The authors only analyzed the difference of NK cell proportion, and should analyze in more detail. The authors should analyze the expression of cytokines using qPCR, ELISA, etc. Additionally, the effect to viral replication should be analyzed. After co-culture with PBMCs, is the HBV replication suppressed?

4. The authors should describe in detail HBV used in this study. Genotype, strain, Accession Number, etc.

5. When using PBMCs, were the cells from each donor pooled or not? Please describe for clarity.

6. According to the Materials and Methods, HepG2-NTCP cells and PBMCs were co-cultured with the same numbers. How about NK cells? When HepG2-NTCP cells and NK cells were co-cultured, was the number of both cells same? If so, the absolute number of NK cells differed between the co-culture with PBMCs and the co-culture with NK cells in Fig. 3 experiment, and I consider comparing the proportion of NK cells is not appropriate. If not, how the authors determine the number of NK cells co-cultured with HepG2-NTCP cells? Please describe clearly.

7. How the authors determine the copy number of HBV genome? Did the authors calculate a standard curve using plasmid such as a recombinant plasmid into which the HBV genome had been inserted? And why the number of cellular genomes (PDHB gene) was measured? Did the authors standardize HBV genome copies by the number of cellular genomes? Please describe clearly.

8. In some Figures such as Fig. 1B, bar with error bar indicate median with upper interquartile range, but it is not appropriate, because the length of upper interquartile range and lower interquartile range are not always same. Please plot all results in a box-and-whisker diagram.

9. In Figure 1A, to evaluate HBV replication in HepG2-NTCP and HepaRG cells, cccDNA and/or viral RNA in cells, and viral DNA in supernatants also should be analyzed.

10. In Materials and Methods section, there is no description about experiments using transwells. Please describe clearly.

11. In Discussion section, discuss more about the relationship between NK cells and monocytes. How about in other viral infections?

Reviewer 3 Report

Comments and Suggestions for Authors

The submitted in vitro study demonstrates an antiviral cytokine production by NK cells in response to contact with HBV-infected HepG2-NTCP cells, dependent on monocyte bystander activation.

Line 86: “For co-culture, PBMCs were added at an effector to target ratio of 1:1”. What was the proportion of HBV-infected cells in the target cell population?

Line 187-188: Is the increase of IFN-gamma expressed as a mean or median? Please indicate the statistical error.

FIGURE 1: Please make clear what is the relation between HBV-infected HepG2-NTCP (Fig.1A) and HBV-infected HepG2 (Fig.1B). What was the MOI and the time of HBV infection of HepG2 in Fig.1B?

What was the effect of antiviral cytokine production by NK cells on the proportion of HBV-infected HepG2-NTCP cells and virus production?

Round 2

Reviewer 2 Report

Comments and Suggestions for Authors

Thanks to the authors for revising the manuscript. The manuscript was improved.

Reviewer 3 Report

Comments and Suggestions for Authors

The manuscript improved substantially. I do not have further comments or questions.